# Recent Progress on Fructose Metabolism—Chrebp, Fructolysis, and Polyol Pathway

**DOI:** 10.3390/nu15071778

**Published:** 2023-04-05

**Authors:** Katsumi Iizuka

**Affiliations:** 1Department of Clinical Nutrition, Fujita Health University, Toyoake 470-1192, Japan; katsumi.iizuka@fujita-hu.ac.jp; Tel.: +81-562-93-2320; 2Food and Nutrition Service Department, Fujita Health University Hospital, Toyoake 470-1192, Japan

**Keywords:** ketohexokinase, *Khk*, glucose transporter 5, *Glut5*, aldolase b, *Aldob*, triokinase/FMN cyclase, *Tkfc*, carbohydrate response element-binding protein, *Chrebp*

## Abstract

Excess fructose intake is associated with obesity, fatty liver, tooth decay, cancer, and cardiovascular diseases. Even after the ingestion of fructose, fructose concentration in the portal blood is never high; fructose is further metabolized in the liver, and the blood fructose concentration is 1/100th of the glucose concentration. It was previously thought that fructose was metabolized in the liver and not in the small intestine, but it has been reported that metabolism in the small intestine also plays an important role in fructose metabolism. *Glut5* knockout mice exhibit poor fructose absorption. In addition, endogenous fructose production via the polyol pathway has also received attention; gene deletion of aldose reductase (*Ar*), ketohexokinase (*Khk*), and triokinase (*Tkfc*) has been found to prevent the development of fructose-induced liver lipidosis. Carbohydrate response element-binding protein (Chrebp) regulates the expression of *Glut5*, *Khk*, aldolase b, and *Tkfc*. We review fructose metabolism with a focus on the roles of the glucose-activating transcription factor Chrebp, fructolysis, and the polyol pathway.

## 1. Introduction

Added sugars (free fructose, sucrose, and glucose) is the name given to sugars that are added to a food by the person or manufacturer preparing it. Added sugars are included in several processed foods, such as sugar-sweetened beverages (SSBs), candies and sugars, desserts and sweet snacks, breakfast cereals and bars, coffee and teas, high-fat- milk, and yogurts. The amount of added sugar intake is associated with the risk of obesity, tooth decay, fatty liver, cancer, and cardiovascular diseases [1,2]. The amount of added sugar intake is also associated with all-cause mortality [3]. The association with all-cause mortality was shown to be significant and dose-dependent for added sugars in beverages (soda/fruit drinks, milk-based drinks, juices, and coffee/teas) but not in solids (treats, cereals, toppings, and sauces) [3]. Indeed, a new WHO guideline recommends that adults and children reduce their daily intake of free sugars to less than 10% of their total energy intake. A further reduction to below 5% or approximately 25 grams per day would provide additional health benefits [1]. Thus, reducing the added sugar intake is beneficial for a healthy life.

Among added sugars with detrimental effects, fructose has an important role in the development of metabolic syndrome diseases [4]. Recently, endogenous fructose production via the polyol pathway has received attention [5]. Fructose is metabolized to pyruvate at a much faster rate than glucose and is used for lipogenesis and gluconeogenesis. As fructose activates carbohydrate response element-binding protein (Chrebp, also called Mlxipl), Chrebp contributes to the pathogenesis of fructose-associated obesity and fatty liver [6,7,8,9,10,11].

Former reviews focused on, especially, exogenous fructose metabolism with special reference to intestinal Chrebp [7,10,11]. However, recent studies have also focused on the relationship between the polyol pathway and fructolysis [5,12]. In this review, we will summarize the role of exogenous and endogenous fructose metabolism in regulating lipogenesis, with special refences to Chrebp (Figure 1).

## 2. Exogenous Fructose Metabolism and Chrebp

### 2.1. Fructose, a Potent Inducer of Lipogenesis

Previously, fructose was considered to be metabolized mainly in the liver; however, the plasma fructose levels are much lower than those in the portal vein and gut lumen [13,14]. Recent findings have shown that the intestine also plays an important role in fructose metabolism [15]. Recently, fructose was shown to be converted into glucose derivatives in the intestine [15]. Fructose enters the enterocytes through the fructose transporter Glut5 (Slc2a5, is converted into fructose-1-phosphate by ketohexokinase (Khk, also called fructokinase), and is then metabolized into glyceraldehyde and dihydroxyacetone phosphate (DHAP) by Aldob [4]. Aldob also catalyzes the reversible cleavage of fructose 1,6-bisphosphate (FBP) into glyceraldehyde 3-phosphate (GA3P) and DHAP [4]. Therefore, Aldob plays a key role in fructolysis and gluconeogenesis [4]. Moreover, Tkfc (triokinase and FMN cyclase) catalyzes the ATP-dependent phosphorylation of the trioses D-glyceraldehyde (GA) and dihydroxy-acetone (DHA) and the cyclizing lyase splitting of FAD (flavin adenine dinucleotide) to AMP (adenosine triphosphate) and riboflavin cyclic-4,5-phosphate (cyclic FMN or cFMN). Among these functions, GA kinase constitutes the third reaction of the Hers pathway for fructose metabolism [16]. *Glut5*, *Khk*, *Aldob,* and *Tkfc* are expressed in the intestine [17,18,19]; however, under normal feeding conditions, the fructose intake does not increase the plasma glucose levels because of suppressed gluconeogenesis. Therefore, fructose has a lower glycemic index than glucose. In contrast, intravenous or peritoneal fructose injection causes increased plasma glucose levels [20]. Interestingly, when fructose or glucose is administered, fructose disappears much more rapidly than glucose, which suggests that fructolysis occurs at a much faster rate than glycolysis [20].

Added sugars include free fructose and sucrose. Although fructose does not affect the plasma glucose levels, it potently induces de novo lipogenesis [21,22,23]. In rats, some reports have compared the potency of fructose on Chrebp transcriptional activity with that of glucose [21]. Two weeks of fructose administration increased Chrebp DNA binding by 3.9 times. In humans, some studies have reported the effects of sugar-sweetened beverages on plasma lipid levels and hepatic lipid synthesis. First, a study was performed to investigate the effect of different types of sugars in SSBs on fatty acid synthesis and oxidation in 34 healthy young men with normal body weight. The relative abundance of palmitate (16:0) and the molar fatty acid ratio of palmitate to linoleic acid (16:0 to 18:2) as markers of fatty acid synthesis were increased after a high fructose ingestion (80 g/d) and a medium fructose ingestion (40 g/d) compared with high sucrose ingestion, high glucose ingestion, or baseline levels. The fasting levels of palmitoylcarnitine, an indicator of impaired fatty acid oxidation flux, were significantly increased after high fructose and high sucrose ingestion [22]. Thus, only fructose in SSBs increases fatty acid synthesis, while fructose and sucrose suppress fatty acid oxidation. Another study also aimed to investigate whether sucrose or fructose induces fat synthesis in 94 healthy men [23]. Seven weeks of administration of SSBs containing moderate amounts of fructose, sucrose (fructose–glucose disaccharide), or glucose (80 g/day) showed that the daily intake of beverages sweetened with free fructose and fructose combined with glucose (sucrose) led to a 2-fold increase in basal hepatic palmitate synthesis compared to the control [23]. Conversely, the same amounts of glucose did not change hepatic palmitate synthesis. Among these groups, there were no differences regarding resting energy expenditure, total fat and carbohydrate oxidation, or the nonprotein respiratory quotient [23]. These results suggest that fructose and sucrose induce de novo lipogenesis much more potently than glucose.

### 2.2. Chrebp, a Regulator of Fructose Metabolism

Why does Chrebp regulate fructose uptake and metabolism? Carbohydrate response element-binding protein (Chrebp) has an important role in regulating *Glut5*, *Khk*, *Aldob,* and *Tkfc* expression [18,24,25]. Chrebp contains two nuclear export signals and one nuclear localization signal near the *N*-terminal, proline-rich domains, a basic helix–loop–helix leucine-zipper domain, and a leucine-zipper-like domain. Glucose and fructose activate Chrebp transcriptional activity and thereby glucose-regulated gene expression, such as fatty acid synthase, acetyl-CoA carboxylase, stearoyl CoA desaturase 1, and Elovl6 [26]. In the livers of *Chrebp*^−/−^ mice, *Glut2*, *Khk*, and *Tkfc* mRNA levels were lower than those in the livers of WT mice [24]. The *Glut5* and *Khk* genes contain carbohydrate response elements (ChoREs), that is, a Chrebp binding site, in their promoter regions [18,27]. Chrebp is abundantly expressed in the liver, kidney, intestine, muscle, adipose tissues, adrenal glands, and pancreatic β cells [24,26,28,29,30,31,32,33,34,35,36]. In the intestine, Chrebp also regulates *Glut5*, *Khk*, *Aldob,* and *Tkfc* gene expression [17,18,19]. Chrebp has two isoforms, *Chrebpα* and *Chrebpβ* [31,37,38]. Chrebpβ is much more potent than Chrebpα, and Chrebpα protein induces *Chrebpβ* mRNA expression [31]. The effects of ChREBPα protein on *Chrebpβ* mRNA induction show positive feedback, while Chrebpβ suppresses *Chrebpα* mRNA expression [35,37]. As *Chrebpβ* is a target gene for Chrebpα, *Chrebpβ* expression is positively correlated with Chrebp transcriptional activity [31]. However, some studies have reported that Chrebpβ is dispensable for lipogenesis. *Chrebpβ*-specific knockout mice have been developed. Interestingly, the lack of the *Chrebpβ* gene showed modest effects on gene expression in the adipose tissues and the liver [38]. Consistent with these findings, in mice fed chow and two types of high-fat diets, a lack of *Chrebpβ* had moderate effects on body composition and insulin sensitivity [38]. The lipid profiles of *Chrebpβ*^−/−^ mice were also similar to those of WT mice [38]. These results suggested that Chrebpα rather than Chrebpβ has at least a dominant role in regulating lipid metabolism in the liver [38].

Which metabolites activate Chrebp transcriptional activity is a difficult question to answer. Chrebp activity is regulated by posttranslational modifications such as phosphorylation/dephosphorylation, acetylation, and O-GlcNAcylation [6,39]. Glucagon and AMP suppress Chrebp transcriptional activity through phosphorylation and allosteric changes in Chrebp [26,40,41,42]. In contrast, some activators exist, such as xylulose-5-phosphate (Xu-5-P) and glucose-6-phosphate (G-6-P) [43,44,45,46]. Uyeda et al. reported that Xu-5-P activates Chrebp transcriptional activity via protein phosphatase 2A (PP2A)-mediated Chrebp dephosphorylation [43]. PP2A also activates phosphofructokinase 2 (PFK2) activity via dephosphorylation [47]. PFK2 catalyzes the formation of a significant allosteric regulator, fructose-2,6-bisphosphate (Fru-2,6-P_2_) [48]. Fru-2,6-P_2_ contributes to the rate-determining step of glycolysis, as it activates the enzyme phosphofructokinase 1 in the glycolysis pathway and inhibits fructose-1,6-bisphosphatase 1 in gluconeogenesis [6]. Thus, Xu-5-P regulates both glycolysis and de novo lipogenesis through PFK2 and Chrebp dephosphorylation mediated by PP2A. Moreover, xylulokinase overexpression caused an increase in Chrebp activity by converting xylitol into Xu-5-P [46]. Last, *Tkfc* gene deletion suppressed fructose-mediated Chrebp activation [49]. These results suggest that xylulose-5-phosphate is an activator of Chrebp.

G-6-P is also an activator of Chrebp [44,50]. Chrebp has a binding site for G-6-P, and G-6-P may activate Chrebp through allosteric effects [50]. In *Chrebp*^−/−^ mice, the intracellular G-6-P levels are increased, and G-6-P may also be an activator of Chrebp. Some studies have reported that fructose 2,6-bisphosphate is essential for the glucose-regulated gene transcription of glucose-6-phosphatase and other Chrebp target genes in hepatocytes [51]. However, it is impossible to examine the effects of G6P or Xu-5-P alone at the cellular level, since G-6-P and Xu-5-P are linked to each other, and their levels fluctuate simultaneously.

### 2.3. Chrebp Gene Deletion and Phenotypes

Chrebp has an important role in regulating glucose and lipid metabolism, especially lipogenesis, glycogen synthesis, and gluconeogenesis. Therefore, through the suppression of Chrebp and the reduction in fatty acid content, improvement in fatty liver is expected. *Chrebp* gene deletion protected ob/ob mice from fatty liver and body weight gain [52]. Ob/ob *Chrebp*
^−/−^ mice showed improved insulin sensitivity and liver triglyceride content [52]. However, massive glycogen accumulation was observed in ob/ob *Chrebp*^−/−^ mice. Similarly, the adenoviral delivery of shRNA against *Chrebp* improved insulin resistance and hepatic steatosis in ob/ob mice [53]. In liver-specific *Chrebp*^−/−^ mice, a high-fructose feed did not induce either body weight gain or Chrebp-targeted gene induction [24,54]. Thus, the liver *Chrebp* gene causes the development of fructose-induced fatty liver changes.

A high-fructose-feed in the presence of *Chrebp* gene deletion also caused hepatomegaly due to massive glycogen accumulation [18,54]. Interestingly, liver glycogen accumulation in *Chrebp*^−/−^ mice disappeared during fasting, and liver histology showed neither inflammation nor fibrosis [55]. These responses are different from those of glycogen storage disease. Liver-type pyruvate kinase overexpression rescues liver cell damage and liver glycogen accumulation [55]. Chrebp regulates glucose 6 phosphatase catalytic subunit (*G6pc*) at the transcriptional level [56], but *G6pc* expression is not completely blunted in *Chrebp^−/−^* mice [24]. These results suggest that *Lpk* suppression causes decreased glycolysis instead of an increased gluconeogenic flux [55]. However, importantly, fibrosis due to massive glycogen storage was not observed. Another study reported that *G6pc*^−/−^ mice showed higher Chrebp activity and increased de novo lipogenesis [57,58,59]. Chrebp suppression caused a decrease in the hepatic TG content in *G6pc*^−/−^ mice but accelerated liver glycogen accumulation and lowered the plasma glucose levels [58,59]. These results suggest the essential role of the Chrebp/G6pc couple and lipogenesis in maintaining liver homeostasis by preventing glycogen-induced hepatomegaly [58,59]. However, the degree of inflammation and fibrosis was much less impressive, considering that in humans, type 1 glycogenic disease can lead to cirrhosis of the liver [60]. Interestingly, some groups have reported the protective role of Chrebp in fructose-mediated steatosis [61]. In *Chrebp*^−/−^ mice, a high-fructose diet reduced the levels of molecular chaperones and activated the C/EBP homologous protein-dependent (CCAAT enhancer-binding protein homologous protein (CHOP)-dependent) unfolded protein response [61]. Moreover, a high-fructose diet induced cholesterol synthesis due to sterol regulatory element-binding protein 2 (Srebp2) activation. The liver free cholesterol, but not total cholesterol, was higher in fructose diet-fed *Chrebp*^−/−^ mice [56]. Thus, Chrebp provides hepatoprotection against a high-fructose diet by preventing the overactivation of cholesterol biosynthesis and the subsequent CHOP-mediated, proapoptotic unfolded protein response [61]. It is an interesting idea that Chrebp may protect against fructose-mediated liver damage. However, some questions remain. Why does *Chrebp* gene deletion activate SREBP2 transcriptional activity in high-fructose-feed conditions? Why is the free cholesterol content higher in fructose-fed *Chrebp*^−/−^ mice? Why do fructose-fed *Chrebp*^−/−^ mice not show severe inflammation or fibrosis despite increased glycogen accumulation and free cholesterol content? The relationship between cholesterol metabolism and Chrebp may be unclear. Thus, *Chrebp* gene deletion has both beneficial effects (lowering the liver triacylglycerol content) and harmful effects (increasing the glycogen content and free cholesterol) in relation to fructose toxicity.

As another characteristic, in both global and intestinal *Chrebp*^−/−^ mice, only 7 days of sucrose administration caused cecal enlargement, diarrhea, and body weight loss [17]. The simple fructose administration also caused similar phenotypes in *Chrebp*^−/−^ mice [18,19]. Moreover, sucrose or fructose feeding was lethal to *Chrebp*^−/−^ mice [17,18,19]. Considering that fructose malabsorption does not cause lethality, high-fructose-fed *Chrebp*^−/−^ mice are a model for not only fructose malnutrition but also hereditary fructose intolerance, which is a life-threatening disease. Moreover, intestine-specific *Chrebp*^−/−^ mice showed fructose intolerance. Therefore, intestinal *Chrebp* has an important role in fructose absorption (Figure 2). Moreover, in the liver, fructose is metabolized into glucose and acetyl coA and thereby used for de novo lipogenesis and glucose output. In the liver, Chrebp regulates not only olfactory fructolytic enzymes but also lipogenic and gluconeogenic enzymes at the transcriptional level [24,25]. Indeed, the changes in fructose concentrations after fructose loading in the portal vein are much larger than those in the peripheral vein in sucrose-fed rats [13,62]. Moreover, after intraperitoneal fructose injection, the hepatic fructose content in *Chrebp*^−/−^ mice was much higher than in WT mice. Thus, Chrebp also has an important role in liver fructose metabolism. In the kidney, fructose is reabsorbed from urine [63]. In seminal vesicles, fructose is endogenously synthesized via the polyol pathway and secreted [64]. It is the major carbohydrate source in seminal plasma and is essential for normal sperm motility [65]. In fact, the fructose concentration in the seminal fluid was found to be as high as 15 mM. Chrebp is also expressed in the testis, but *Chrebp*^−/−^ mice are normally fertile. Further investigation will be needed to clarify the relationship between fructose metabolism and Chrebp in these tissues.

### 2.4. Phenotypes Induced by Glut5, Khk, Aldob, and Triokinase Mutations

Glut5 is the primary transporter responsible for the facilitative absorption of fructose. *Glut5* is expressed in the small intestine (duodenum), testis, and kidney to a lesser extent. Sugars and cAMP induce *Glut5* expression [66,67]. Chrebp binding sites are found in the *Glut5* promoter, and *Chrebp* gene deletion causes decreased *Glut5* expression [17,18,19]. Moreover, the posttranscriptional regulation of human *GLUT5* by fructose involves increases in mRNA stability mediated by the cAMP pathway and Paip2 (PABP-interacting protein 2) binding [68]. Human *GLUT5* expression can also be induced by thyroid hormone (triiodothyronine, T3) and glucocorticoids through the activation of the thyroid hormone receptor (THR) and the glucocorticoid receptor (GR), respectively [69]. Recently, LXR was reported to induce human *GLUT5* expression [70]. Glucose and thyroid hormone, as well as glucose and cAMP, coregulate the expression of the intestinal fructose transporter *GLUT5* [69]. The human *GLUT5* gene was cloned in a patient with isolated fructose malabsorption; however, the human *GLUT5* mutation did not contribute to acquired fructose malabsorption [71]. *Glut5*^−/−^ mice fed a normal diet had normal blood pressure and displayed a normal weight gain. In contrast, a high-fructose diet caused hypotension and massive dilatation of the cecum and colon due to fructose malabsorption [72]. Moreover, *Glut5*^−/−^ mice exhibited no facilitative fructose transport and no compensatory increases in the activity and expression of *Sglt1* and other glucose transporters [72]. Consistent with these findings, fructose could not upregulate *Glut5* in *Khk*^−/−^ mice. These results are consistent with the finding that glucose derivatives induce *Glut5* gene expression via Chrebp activation.

Khk is an enzyme that catalyzes the phosphorylation of fructose to produce fructose-1-phosphate [73,74]. A human *KHK* gene mutation causes only essential fructosuria [73,74,75,76]. In patients with *Khk* gene mutations, ingested fructose is partly (10–20%) excreted in urine, and the rest is slowly metabolized by an alternative pathway, namely, it is converted into fructose-6-phosphate by hexokinase in the adipose tissue and muscle [73,74,75,76]. The mode of inheritance is autosomal recessive, and the homozygote frequency has been estimated at 1:130,000. Interestingly, an intravenous fructose injection causes no hepatic metabolic changes (phosphomonoester, ATP, and Pi) in patients with essential fructosuria [77]. Thus, essential fructosuria is a harmless anomaly characterized by the appearance of fructose in the urine after the intake of fructose-containing food.

In mice, *Khk* gene suppression may also be beneficial for liver steatosis and hyperuricemia. However, the suppression of the polyol pathway does not improve glucose clearance and partly decreases glucose clearance. Moreover, after glucose feeding, Khk-A/C suppression did not improve either insulin tolerance or HbA1c levels [78]. However, excess fructose intake will increase the plasma fructose levels and the accumulation of advanced glycation end products (AGEs) because fructose-derived AGEs may be involved via the Maillard reaction, and fructose is 10 times more reactive than glucose, although the plasma fructose concentration is only 1% of that of glucose [79]. If the plasma fructose levels continue to increase, fructose-derived AGEs are increased and may cause cell damage due to intracellular AGEs in vascular cells [79]. Although Khk deficiency is a benign disease that causes fructosuria, it is expected that the continuous intake of excess fructose can lead to hyperfructosemia, a health problem caused by protein glycation, as seen in diabetes mellitus patients.

Aldob plays a key role in both glycolysis and gluconeogenesis. A human *ALDOB* gene mutation causes hereditary fructose intolerance [75,80,81]. Fructose intolerance presents with abdominal pain, nausea, and hypoglycemia symptoms, as well as shock-like syndrome after fructose ingestion [80]. The incidence of hereditary fructose intolerance, which is an autosomal recessive disease, is estimated to be 1 in 20,000 to 30,000 individuals each year worldwide. A lack of *Aldob* results in the accumulation of fructose-1-phosphate (F-1-P) in liver and renal cells. F-1-P is toxic and causes cell damage. As *Aldob* regulates fructolysis (energy production) and gluconeogenesis, *Aldob* gene deletion causes decreased cellular energy, low blood sugar levels, renal tubular acidosis, and severe liver dysfunction. Thus, in hereditary fructose intolerance, fructose may provoke prompt gastrointestinal discomfort and hypoglycemia upon ingestion. Fructose-fed *Aldob*^−/−^ mice showed similar phenotypes to humans [80]. Interestingly, some studies reported that *Aldob* gene depletion promotes hepatocellular carcinogenesis by activating insulin receptor signaling and lipogenesis [81]. Another group also reported that hepatic *Aldob* gene deletion activates Akt and promotes hepatocellular carcinogenesis by destabilizing the Aldob/Akt/PP2A protein complex. These results are consistent with the inverse correlation between *Aldob* and p-Akt expression in human HCC tissues [82].

Triokinase/FMN cyclase is an enzyme that in humans is encoded by the *TKFC* gene [83]. This is a homodimeric protein with subunits containing two domains (K and L domains or *N*-terminal and *C*-terminal). Both domains are needed for triokinase activity, while the L domain suffices for FMN cyclase activity [83]. Triokinase and FMN cyclase (TKFC) encode a bifunctional enzyme involved in fructose metabolism through its glyceraldehyde kinase activity and in the generation of riboflavin cyclic 4′,5′-phosphate (cyclic FMN) [83].

Triokinase and FMN cyclase deficiency syndrome (TKFCD) is a multisystem disease. In addition to cataracts and developmental delays of varying severity, other features include liver dysfunction, microcytic anemia, and cerebellar hypoplasia [84]. Fatal cardiomyopathy with lactic acidosis has been observed. These phenotypes are due to the biallelic mutations c.1628G>T or c.1333G>A in the *C*-terminal region. In contrast, biallelic mutations of the *N*-terminal domain caused only hypotrichosis [85]. A patient with both c.574G>C and c.682C>T mutations showed hypotrichosis, while the patient’s father and mother with heterogenous mutations of either c.574G>C or c.682C>T did not show hypotrichosis [85]. These results indicate that TKFC functions as a homodimer and that patients with genetic mutations in the *C*-terminal domain show more severe phenotypes.

In *Tkfc*^−/−^ mice, fructose diets cause diarrhea and cecal enlargement, which is consistent with the fructose malabsorption phenotypes observed in *Glut5*^−/−^ and *Chrebp*^−/−^ mice [50]. High fructose diet-induced hepatic lipogenesis and steatosis were effectively reduced by *Tkfc* knockdown [50]. Interestingly, triokinase gene deletion suppressed Chrebp transcriptional activity. Moreover, the highly prevalent human variant Ala185Thr-TKFC has been reported to be ‘null’ for fructose metabolism, since Ala185-TKFC rescues the mouse TKFC-deficient phenotype, whereas Ala185Thr-TKFC does not. In contrast, Thr185-TKFC is actually fully active as a triokinase/FMN cyclase, and a yeast growth assay revealed that Ala185-TKFC and Ala185Thr-TKFC overexpression rescued growth in Δdak1 yeast cells. These results suggest that human Thr185-TKFC cannot be considered a ‘null’ variant in terms of triokinase activity [86].

### 2.5. Comparison between Chrebp^−/−^ and Fructose-Regulated Gene Knockout Mice

Chrebp regulates *Glut5, Khk, Aldob*, and *Tkfc* expression [17,18,19,24], and *Chrebp* gene deletion is predicted to worsen fructose insufficiency. Gene deletion of ketohexokinase (*Khk*), an enzyme upstream of Aldob, is sufficient to prevent hypoglycemia and liver and intestinal injury associated with feeding *Aldob*^−/−^ mice fructose [80]. Moreover, only *Khkc* suppression rescued the phenotypes of hereditary fructose intolerance in fructose-fed *Aldob*^−/−^ mice [80]. Considering that the phenotypes associated with alterations of downstream enzymes (*Aldob* and *Tkfc*) are more life-threatening than those associated with *Glut5* and *Khk* alteration, decreased *Khk* and *Glut5* expression may alleviate fructose insufficiency due to Aldob or *Tkfc* suppression. However, fructose-fed *Chrebp*^−/−^ mice showed lethality, and the effects of *Khk* and *Glut5* may be limited. Thus, as *Chrebp* gene deletion suppresses multiple fructolytic steps, *Chrebp* gene deletion led to a phenotype characterized by both fructose malabsorption and fructose intolerance.

### 2.6. Other Transcription Factors (ATF3, Srebp1c, AhR) and Fructose

Activating transcription factor 3 (ATF3) is a member of the ATF/cAMP responsive element-binding protein family of transcription factors with the basic region leucine zipper (bZip) DNA-binding domain [87]. In combination with a homodimer and various heterodimers with other bZip proteins, such as ATF2, c-Jun, JunB, and JunD, ATF3 can function as a transcriptional activator or repressor [87]. Some studies have reported the role of ATF3 in the pathogenesis of metabolic syndrome induced by a high-fructose diet. ATF3 KO in mice increased the serum levels of glucose, insulin, triglycerides, and inflammation markers (tumor necrosis factor-alpha and intercellular adhesion molecule-1), with increased visceral adiposity [87]. ATF3 gene deletion did not ameliorate the metabolic parameters or inflammatory cytokines. The authors concluded that ATF3 deficiency is involved in the pathogenesis of metabolic syndrome. However, the changes in ATF3 activity in metabolic syndrome remain unclear [87]. Srebp1c is also a transcription factor that regulates lipogenic gene expression in the liver [88]. Fructose induces Srebp1c mRNA levels in a time-dependent manner, but the time course changes in Srebp1c mRNA induced by fructose are different from those induced by glucose. In insulin-deficient streptozotocin-injected mice, fructose also increases *Srebp1c* mRNA levels [88]. These results suggest that fructose also activates the transcription factor sterol regulatory element-binding protein 1c (Srebp-1c), independent of insulin action [88]. Pgc-1β is a coactivator of SREBP1c [88]. The administration of antisense oligonucleotide against *Pgc-1β* mRNA improved the phenotypes (hepatic lipid synthesis, hepatic glucose production) caused by a high-fructose diet, which was due to reduced *Srebp1c* expression [89]. Another study reported that feeding fructose-containing water caused an increase in *Srebp1c* mRNA levels in the hypothalamus of Wistar rats [90]. Thus, the relationship between Srebp1c and fructose intake is observed in several tissues. Aryl hydroxycarbon receptor (AhR) functions primarily as a sensor of xenobiotic chemicals and as a regulator of enzymes such as cytochrome P450s that metabolize these chemicals [91]. Interestingly, some studies have reported that xenobiotic signaling pathways are interlinked with fructose consumption. Fructose suppressed AhR signaling by modulating the expression of transcription factors (AhR nuclear translocator) and upstream regulators (Ncor2 and Rb1) [91]. As a result, fructose suppressed biotransformation gene expression (*Cyp1a2*, *Ugt1a1*, *Nqo1*, and *Gsts*). Considering that Chrebp is a negative regulator of ARNT/HIF-1β gene expression in pancreatic islet β-cells [92], fructose may also inhibit AhR expression via Chrebp activation in the liver.

## 3. Endogenous Fructose Production

### 3.1. The Polyol Pathway and Endogenous Fructose Production

When the glucose levels become greatly elevated, other pathways are upregulated. These pathways include the glycation pathway, the hexosamine pathway, the protein kinase C pathway, the alpha-ketoaldehyde pathway, and the sorbitol pathway [93,94,95].

The polyol pathway converts excess glucose into sorbitol and fructose. In diabetic conditions, the polyol pathway is activated and accelerates diabetic microvascular complications such as diabetic neuropathy [93,94,95]. Under normal conditions, the polyol pathway has a low flux because Ar has higher K_M_ for glucose, and the conversion of glucose into sorbitol requires high glucose concentrations. The polyol pathway contains two enzymes, aldose reductase and sorbitol dehydrogenase [93,94,95].

### 3.2. Findings in Sorbitol Dehydrogenase Knockout Mice

Sorbitol dehydrogenase is expressed in only limited tissues, such as the ovaries, seminal vesicles, liver, kidney, and lens, and is not expressed in the retina or Schwann cells. Recently, Sano et al. reported that gene deletion of sorbitol dehydrogenase (*Sord*) caused a decrease in glucose-induced Chrebp nuclear translocation in the liver, while the effect of fructose on Chrebp nuclear translocation was preserved in *Sord*^−/−^ mice [96]. *Sord*^−/−^ mice are glucose-intolerant, which reveals that the glucose flux partly bypasses the glycolytic pathway into polyol pathways [96]. Despite these interesting results, we should interpret the data cautiously because much of the fructose injected is converted into glucose and its derivatives in the intestine. Therefore, the effect of glucose and fructose on Chrebp nuclear translocation using isolated primary hepatocytes should be examined. Moreover, although the nuclear translocation of Chrebp was suppressed, the triacylglycerol levels were unchanged, and the authors did not show the effect of Chrebp-targeted gene expression [96]. These results suggested that glucose fluctuation by the polyol pathway is transient and that the effect of the polyol pathway on Chrebp activation may also be transient. However, persistent hyperglycemia may increase the contribution of the polyol pathway to Chrebp activation because persistent hyperglycemia causes increased NADPH production via the pentose phosphate shunt (Figure 1).

### 3.3. Findings in Aldose Reductase Knockout Mice

Aldose reductase (Ar) is a rate-limiting enzyme in the polyol pathway. When feeding a glucose solution, Ar protein density was increased 1.5-fold, but *Ar* (Akr1b1) mRNA levels were unchanged [97]. Consistently, the sorbitol and fructose levels also increased. In *Ar*^−/−^ mice, glucose solution feeding prevented body weight gain and hepatic lipid accumulation compared with WT mice [97]. However, glucose feeding did not affect insulin tolerance, and the plasma glucose levels were also higher. These results suggest that Ar inhibition does not improve glucose tolerance.

In contrast, the relationship between liver disease and Ar is controversial. Some authors reported that Ar protein expression was significantly higher in genetically obese mice with leptin receptor mutation, db/db mice, fed a methionine–choline-deficient (MCD) diet than in mice fed a control diet [98]. *Ar* gene silencing decreased the levels of serum alanine aminotransferase and hepatic lipoperoxides, reduced the mRNA and protein expression of hepatic cytochrome P450 2E1 (CYP2E1), and decreased the mRNA expression of proinflammatory tumor necrosis factor-a (TNF-a) and interleukin-6 (IL-6) [98]. A relationship between alcoholic fatty liver and aldose reductase has been reported. Alcohol ingestion induces *Ar* mRNA expression in the liver. Zopolrestat, an AR-specific inhibitor, improved ethanol-induced steatosis and hepatic oxidative stress by suppressing adenosine monophosphate-activated protein kinase (AMPK) activation and Srebp1c expression in C57BL/6J mice and HepG2 cells [99,100]. Inconsistent with these data, others have reported that epalrestat (EPS), an aldose reductase inhibitor, increased oxidative stress, as indicated by the increased expression of manganese superoxide dismutase (heme oxygenase-1 and NAD(P)H quinone oxidoreductase-1)-induced inflammation, and the infiltration of inflammatory cells and induced the expression of tumor necrosis factor-alpha, CD11b, and CD11c, leading to fibrosis in mouse liver [101]. Further investigation should be conducted to clarify the relationship between Ar and liver dysfunction. Moreover, the relationship between Chrebp and Ar has not been reported. While chronic alcohol consumption induces Chrebp activity through dephosphorylation [102], binge drinking induces Chrebp through acetylation [103]. Since alcohol ingestion induces *Ar* mRNA expression in the liver, Ar inhibition may also suppress Chrebp activation. The relationship between Ar and liver diseases remains unclear.

### 3.4. The Polyol Pathway and Endogenous Fructose Metabolism

Some groups have mentioned that the polyol pathway in the kidney plays an important role in kidney disease. Their hypothesis is based on the activation of the polyol pathway, as evidenced by high levels of aldose reductase, sorbitol, and endogenous fructose. The plasma fructose levels (10 μM) are much lower than the plasma glucose levels (5~10 mM) [104]. Considering that fructose is converted into glucose in the intestine and that the portal fructose levels are much lower than the glucose levels, hepatic endogenous fructose production via the polyol pathway may be an attractive candidate as the causative pathway of obesity, fatty liver, and hyperuricemia in patients consuming excess sugar-sweetened beverages. However, the role of the polyol pathway in the liver is not yet well studied [5]. Lanaspa et al. reported interesting findings, showing that *Khk* gene deletion prevented glucose-induced body weight gain and hepatic lipid accumulation [97]. Despite Khk suppression in *Khk*^−/−^ mice, the hepatic fructose content in *Khk*^−/−^ mice fed glucose were similar to those in WT mice. However, the fatty acid synthase and ATP citrate lyase levels in *Khk*^−/−^ mice were similar to those in WT mice. These results are not consistent with those of another paper showing that uric acid activated ChREBP through Khk induction [105]. Moreover, gene deletion of aldose reductase also prevented glucose-induced body weight gain and fatty liver development [89]. The effects of *Ar* gene suppression on body weight gain and hepatic liver accumulation seem to be superior to those of *Khk* gene suppression. Considering that Ar is a first step in the polyol pathway, *Ar* gene suppression may be more effective as anti-obesity therapy than Khk suppression.

Sanchez-Lozada et al. also reported that uric acid activates Ar [105]. However, the sorbitol and fructose content in HepG2 cells also increased despite increased Khk activity. They also reported that uric acid stimulates Khk activity and fructose metabolism through Chrebp activation. Moreover, they speculated that uric acid could increase Chrebp activity through Khk activation. However, the increase in Khk levels was very low, so the contribution of uric acid to Chrebp-mediated *Khk* induction may be low, and its clinical significance seems to be minimal [105]. Taken together with the findings on the sorbitol and fructose content, these results suggest that the potency of Khk activated by uric acid is lower than that of aldose reductase in HepG2 cells, which is why sorbitol and fructose accumulate in HepG2 cells [97]. If the authors had used primary hepatocytes, their results could have been different and more convincing, because Chrebp activation in HepG2 cells is much weaker than in primary hepatocytes. In obese patients, overproduction and hyposecretion of uric acid vary. Moreover, uric acid levels are generally decreased during the development of diabetes mellitus, and this mechanism may be active only in the glucose intolerance stage, when urinary glucose and uric acid excretion is not increased.

Considering that aldose reductase, sorbitol dehydrogenase, and ketohexokinase are highly expressed in the liver, it is an interesting idea that uric acid promotes the polyol pathway and endogenous fructose production. Further investigation will be needed to clarify the role of the hepatic polyol pathway and endogenous fructose production in hyperuricemia.

### 3.5. Effects of Suppressing the Polyol Pathway on Glucose Tolerance

As the polyol pathway is a bypass to deal with excess glucose, the polyol pathway can be described as a pathway that allows excess incoming glucose to escape through the bypass, so that glucose that is not fully taken up remains in the blood and hyperglycemia can occur. Indeed, *Sord*^−/−^ mice showed glucose intolerance. Therefore, it is very important to note that the endogenous fructose production pathway via the polyol pathway does not necessarily improve glucose tolerance.

## 4. Conclusions

The recent findings on exogenous and endogenous fructose metabolism are very interesting topics. Although the relationship between fructose intake and various metabolic diseases has been reported, the fact that the small intestine acts as a fructose barrier and that fatty liver occurs despite a low fructose concentration in the portal vein after fructose loading suggests the role of metabolites such as pyruvate, which is metabolized from fructose in the small intestine. It may be necessary to investigate the role of metabolites such as pyruvate that are generated from fructose in the small intestine. Furthermore, the role of the endogenous fructose pathways, including the polyol pathway, may also be important when excessive glucose is ingested. The relationship between fructose intake and metabolic diseases is still largely unknown. Further research is needed to elucidate the molecular mechanisms of fructose intake-induced diseases, as this is important not only for drug development but also for patient education.

## Figures and Tables

**Figure 1 nutrients-15-01778-f001:**
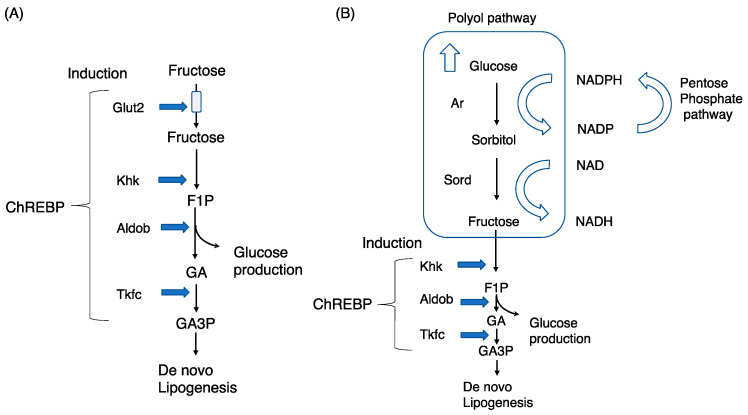
Chrebp regulates exogenous and endogenous fructose metabolism in the liver. (**A**) Exogenous fructose metabolism: fructose is transported into the cytosol via Glut2 (liver) and is then converted into fructose-1-phosphate (F1P), glyceraldehyde (GA), and GA3P via ketohexokinase (Khk), aldolase b (Aldob), and triokinase/FMN cyclase (Tkfc), respectively. (**B**) Endogenous fructose metabolism: the polyol pathway comprises aldose reductase (Ar) and sorbitol dehydrogenase (Sord). Excess glucose is converted into sorbitol and fructose via Ar and Sord, respectively. Fructose is similarly converted into F1P, GA, and GA3P.

**Figure 2 nutrients-15-01778-f002:**
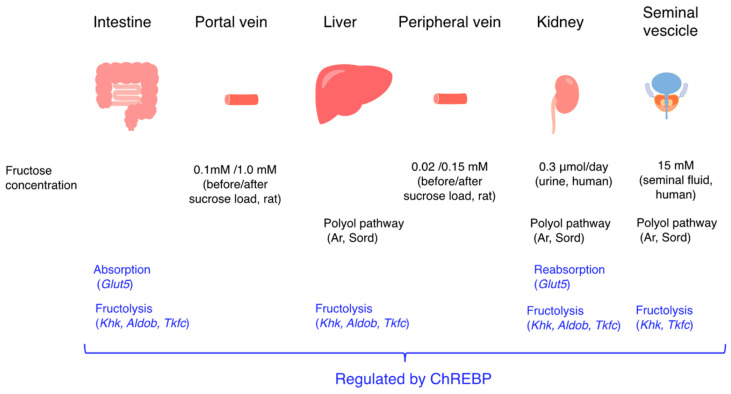
Fructose promotes intestinal and liver fructose metabolism through Chrebp activation. Chrebp regulates *Glut5*, *Khk*, and *Aldob* expression and then promotes intestinal fructose absorption and hepatic fructose metabolism (de novo lipogenesis and glucose production). The changes in fructose concentration in the portal vein and in the peripheral vein after sucrose loading are also indicated. Fructose concentrations in human urine and seminal fluid are also indicated. In the seminal vesicles, Aldob was not detected.

## Data Availability

Not applicable.

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
