# Peer review of "Recent Progress on Fructose Metabolism—Chrebp, Fructolysis, and Polyol Pathway"

_nutrients, 2023, doi:10.3390/nu15071778_

Round 1
Reviewer 1 Report
The manuscript by Katsumi Iizuka summarizes exogenous and endogenous fructose metabolism in intestine and liver. The author also describes the role of ChREBP on Glut5, KHK, aldolase B and TKFC which are the main enzymes involved in fructose metabolism. This manuscript includes recent updates in the research on the fructose-associated obesity and fatty liver disease.
Overall, while the topic of manuscript is interesting, there are several concerns that should be addressed.
Major concerns
1. The title should be more clear. The author focuses on the role of ChREBP in regulating fructose metabolism. Other transcription factors that affect the expression of fructose metabolizing genes should be better discussed.
2. It has been shown that small intestine acts as a shield for fructose induced metabolic derangements. Although fructose metabolism in the intestine is well documented by Dr. Rabinowitz lab, intestine can only metabolize a small amount of fructose about 3-5 g. Higher fructose load reaches the liver and is primary metabolized there. This point should be better stated in the manuscript. Currently it overestimates the contribution of the intestines.
3. Glut5 is the main fructose transporter in the intestine, but not in the liver. As liver is one of major organs for the fructose metabolism, the author should discuss tissue specific fructose metabolism. Please see PMID: 33057854
4. Figure 1 should list either protein or gene symbols for consistency.
5. Figure 2 is too simplistic and not informative. This is an excellent chance to highlight tissue specific fructose metabolism.
Minor concerns
Both terms “fructokinase” and “ketohexokinase” are used in the manuscript interchangeably. The aut
Author Response
Reviewer 1
Q1: Overall, while the topic of manuscript is interesting, there are several concerns that should be addressed.
A1: Thank you very much for your positive comments. I followed your advise and thoroughly rewrote the manuscript.
Q2: The title should be much more clear. The author focuses on the role of ChREBP in regulating fructose metabolism. Other transcription factors that affect the expression of fructose metabolizing genes should be better discussed.
A2: Thank you very much for your excellenet suggestion. According to your suggestions, we added one section (2.5. Other transcription factors (ATF3, Srebp1c, AhR) and fructose)
Q3: It has been shown that small intestine acts as a shield for fructose induced metabolic derangements. Although fructose metabolism in the intestine is well documented by Dr. Rabinowitz lab, intestine can only metabolize a small amount of fructose about 3-5 g. Higher fructose load reaches the liver and is primary metabolized there. This point should be better stated in the manuscript. Currently it overestimates the contribution of the intestines.
A3: According to your suggestions, I emphasized the role of liver, too.
Q4: Glut5 is the main fructose transporter in the intestine, but not in the liver. As liver is one of major organs for the fructose metabolism, the author should discuss tissue specific fructose metabolism. Please see PMID: 33057854
A4: According to your suggestion, we have unified the notation with the GENE symbol. As you mentioned, GLUT2 is involved in uptake in the liver. In the small intestine, GLUT5 is involved in uptake from the lumen and GLUT2 is involved in secretion into the portal vein. In accordance with the above, we correct the notation of transporters in the liver.
Q5: Figure 1 should list either protein or gene symbols for consistency.
A5: According to your suggestions, I corrected Figure 1 for consistency.
Q6: Figure 2 is too simplistic and not informative. This is an excellent chance to highlight tissue specific fructose metabolism.
A6: According to your advise, I added some information about fructose concentration and other tissues such as kidney and seminal vesicle.
Q7: Both terms “fructokinase” and “ketohexokinase” are used in the manuscript interchangeably.
Aï¼—: We will follow your suggestions and write ketohexokinase in a unified manner.
Reviewer 2 Report
In this review, the author elaborates on the effects of the carbohydrate response element binding protein (ChREBP) over genes of (i) the absorption and metabolism of exogenous fructose and (ii) the pathway for endogenous fructose production.
Concerning absorption and metabolism of exogenous fructose and ChREBP, the author first discusses that fructose is stronger than glucose as an inducer of lipogenesis, and then the role of ChREBP as regulator of metabolism, through effects on four genes: Glut5 (currently Slc2a5), Khk, Aldob and Tkfc. These genes encode, respectively, the transporter responsible for fructose absorption in the small intestine, and the three enzymes catalyzing fructose metabolism in intestine and liver, i.e. fructokinase, aldolase B and triokinase. The part devoted to exogenous fructose metabolism ends with an interesting discussion of the phenotypes of mutations in the Chrebp gene (currently Mlxipl) as well as in Glut5, Khk, Aldob and Tkfc, either in knockout mice or in human patients.
Concerning endogenous fructose metabolism, the author first discusses “The polyol pathway and endogenous fructose production”, and afterwards “The polyol pathway and endogenous fructose metabolism”. The relationship between these two sections, the underlying line of thought and the resulting message in this part, are difficult to follow.
Altogether, the manuscript is interesting and generally well written, although the part concerning endogenous fructose metabolism is less satisfactory.
The bibliography cited is very current and relevant. However, there are a few essential references missing and some imprecision in the way of citing. After revision to solve this and other issues found, the manuscript would represent a significant addition to the very busy topic of fructose metabolism.
MAJOR ISSUES
1.- After searching the literature on fructose metabolism for reviews (with fructose as MeSH Major Topic and fructose as title word), 181 reviews were recovered in the period 2012-2023, seven of them in 2022-2023. Two of these reviews, dated 2017 and 2018 (Pubmed IDs 28241431 and 30158026) deal with a topic similar to Dr. Iizuka’s manuscript, i.e. ChREBP role in intestinal and/or liver fructose absorption and metabolism. None of them is cited in the current manuscript. One of these (Pubmed ID 28241431) is also authored by Dr. Iizuka in Nutrients. He also has published another ChREBP review in 2021 although with a different approach (reference 7 in the current manuscript). Therefore, both not cited reviews (Pubmed IDs 28241431 and 30158026), should be cited in the Introduction and the need for a new review should be justified, with emphasis on new developments or aspects not adequately covered in the earlier reviews.
2.- Triokinase is not well represented in several parts of the manuscript, as detailed below, in points 2a, 2b and 2c.
2a.- In line 64, It is stated that “In the intestine, Glut5, Khk, and Aldob are abundantly expressed [11-13]”. This seems to imply that Tkfc expression would be much lower than the other three genes. However, by checking expression data for mouse genes in the NCBI Gene database (https://www.ncbi.nlm.nih.gov/gene; gene IDs,), one can see the following data from the Mouse ENCODE transcriptome (Pubmed ID: 25409824) for adult duodenum: Tkfc (gene ID 225913) 97.7 rpkm; Glut5 (current official name Slc2a5)(gene ID 56485) 45.4 rpkm; Khk (gene ID 16548) 161.7 rpkm; Aldob (gene ID 230163) 341.1 rpkm. Tkfc expression is clearly stronger in duodenum than in many other mouse tissues. Therefore, I do not see a clear rationale to exclude Tkfc in the sentence of line 64. Tkfc should be added to that sentence and reference 10 (that contains data for Tkfc expression —see page 358, right column, end of first paragraph; and Figure S5E) should be included in the parenthesis, together with Pubmed ID: 25409824 (Mouse ENCODE transcriptome). The author may consider deleting or substituting the word “abundantly” from line-64 sentence. If rather than mice, one considers human gene data, the conclusion would be similar.
2b.- In Figure 2, Tkfc is not shown as responding to ChREBP, while the contrary is stated in Figure 1 and in the text (lines 98-99 and lines 108-109; according to data in references [18,31]). Therefore, the Tkfc gene should be included in Figure 2 as being activated by ChREBP.
2c.- The paragraph devoted to Tkfc phenotypes (lines 255-272) contains several important imprecisions that should be corrected.
–In line 255, DAK gene is an outdated symbol that must be substituted by TKFC, the official gene symbol assigned by the HUGO Gene Nomenclature Committee as proposed in the key paper pubmed ID 24569995. This key paper contains full data for structure and activities of triokinase/FMN cyclase, and it should be cited in line 255-256 instead of reference [71].
–Reference [71] should be maintained in line 261, for the description of the TKFCD syndrome, just as it appears now.
–In line 258 remove “through an FMN lyase domain” since this domain name, although used in reference [71], is misleading. It implies erroneously that there is a lyase and a kinase domain. To be specific about domains, I suggest adding this description in line 258 after “…(cyclic FMN).”: “This is a homodimeric protein with subunits containing each two domains (K and L, or N-terminal and C-terminal). Both domains are needed for triokinase activity while the L domain suffices for FMN cyclase activity (Pubmed ID 24569995).”
–Concerning the TKFCD syndrome (line 259), you should consider that it is caused by point mutations in the L (or C-terminal) domain that inactivate the triokinase activity (without data on FMN cyclase) [71]. This is important because mutations in the K (or N-terminal) domain, which also inactivate triokinase and most of FMN cyclase activity, cause a benign hair phenotype (Pubmed ID 32790068).
–In lines 264 and 265, reference [72] should be changed to [44] (this is cited in lines 138, 140 and 141), since [44] and [72] are the same duplicated reference.
–In line 269, after “Contrary to this result,” insert the following: “Thr185-TKFC is actually fully active as triokinase/FMN cyclase (Pubmed ID 24569995), and a yeast …” etc.
(Please, note as article Pubmed ID 24569995) is the only full molecular and enzymatic characterization of triokinase/FMN cyclase published and actually it is Thr185-TKFC). This is clearly stated in reference [73].
3.- One of the main points of the review manuscript is the role of ChREBP as regulator of fructose absorption and metabolism. The evidence sustaining this is summarized in lines 97-108, where references [18,19] are cited in relation to effects on Glut5, Khk, Aldob and Tkfc genes in liver, and references [12,13,31] for effects in intestine. However, the use of the references is imprecise, and matching the references with the genes is difficult.
–In line 99, [18,19] together contain data for the four genes, but [18] only for Khk and Tkfc, and [19] for Glut5, Khk and Aldob.
–In line 105, [18] is cited for data about Glut5, Khk and Tkfc, but actually it does not contain Glut5 data.
–In line 109, [12,13,31] are cited for data about the four genes, but only [31] contains data for all of them.
A more precise pattern of cites would be helpful to the reader in this part of the manuscript, to facilitate access to the original data in which the main point of the review relies.
4.- The author should improve the part devoted to endogenous fructose metabolism. The relationship and differences between sections 3.1 and 3.2 are difficult to grasp. They should be better defined, with a more clear line of thought and resulting message. This could be probably improved by presenting a more structured content.
5.- Gene names and symbols should be carefully revised throughout the manuscript to ensure they are shown in italics, in lower case or in capital letters as required.
6.- An abbreviation list is missing. For instance: SSBs, DHAP, G6P, X5P, TKFCD, HFrD, AR, AMPK, EPS, REE, CHO, ChoREs, MCD, Ho-1, Nqo1, TAG, etc (of course, gene symbols are not to be included in the list). In cases when the abbreviation is used only once, remove the abbreviation. Otherwise, please use them consistently throughout the manuscript.
MINOR ISSUES
—As stated in my general comments at the beginning of this report, there are two genes which now have changed names and symbols: Glut5 is now officially Slc2a5, and Chrebp is now Mlxipl. I agree that in these cases it is better to maintain the more common old names, but the new names could be mentioned in parenthesis at their first use.
—Line 40, it reads “Fructose is metabolized faster and…”. Please state “faster than… what?
—Figure 1A, The meaning of “Activation” in the top of the figure should be explained, modified or deleted, as necessary.
—Line 97, it reads “…fructose regulates fructose…”. This sentence, if indeed meant, needs an explanation. However, perhaps it just should read “…GhREBP regulates fructose…” ?
—Lines 112-113, it reads “As Chrebpb is a target gene for Chrebpa,…”; should it read “As Chrebpb is a target gene for ChREBPa, …”? or else “As Chrebpb is a target gene for Chrebpa,…”? (see major point 5).
—Lines 286-289, I do not believe that the glycation pathway can be considered as a pathway to handle glucose effectively. It rather is a toxic consequence of high glucose levels. The sentence would be acceptable if you delete “to handle glucose effectively”.
—Line 293, “higher km activity” is incorrect in the spelling of KM and in the use of the term “activity”. Substitute just by “higher KM”.
—Line 304, Khk-A/C, please explain that A and C are Khk isoenzymes.
—Line 306, please explain what are db/db mice.
—Line 328, Sord should be Sord (see major point 5).
—Line 369, substitute “activated” by “activation”.
—Lines 374-375, the meaning of “consistent” here is unclear. Please, rephrase.
—Lines 399-400, this sentence needs references.
—Line 417, “since not only is fructose taken in but also excessive glucose is ingested”, mention to what situation is this referred to.
Author Response
(Reviewer 2)
Q: In this review, the author elaborates on the effects of the carbohydrate response element binding protein (ChREBP) over genes of (i) the absorption and metabolism of exogenous fructose and (ii) the pathway for endogenous fructose production.
Altogether, the manuscript is interesting and generally well written, although the part concerning endogenous fructose metabolism is less satisfactory.
After revision to solve this and other issues found, the manuscript would represent a significant addition to the very busy topic of fructose metabolism.
A: Thank you very much for your precise suggestions. I followed your advise and thoroughly rewrote the manuscript.
MAJOR ISSUES
Q1: both not cited reviews (Pubmed IDs 28241431 and 30158026), should be cited in the Introduction and the need for a new review should be justified, with emphasis on new developments or aspects not adequately covered in the earlier reviews.
A1: Thank you very much for your comments. Accoreding to your suggestions, I cited two reviews. Recent finding on fructose metabolism on polyol pathway and triokinase are also accumulated.
Q2.- Triokinase is not well represented in several parts of the manuscript, as detailed below, in points 2a, 2b and 2c.
A2: This is a very important suggestion. Thank you.
Q: 2a.- In line 64, It is stated that “In the intestine, Glut5, Khk, and Aldob are abundantly expressed [11-13]”. This seems to imply that Tkfc expression would be much lower than the other three genes. However, by checking expression data for mouse genes in the NCBI Gene database (https://www.ncbi.nlm.nih.gov/gene; gene IDs,), one can see the following data from the Mouse ENCODE transcriptome (Pubmed ID: 25409824) for adult duodenum: Tkfc (gene ID 225913) 97.7 rpkm; Glut5 (current official name Slc2a5)(gene ID 56485) 45.4 rpkm; Khk (gene ID 16548) 161.7 rpkm; Aldob (gene ID 230163) 341.1 rpkm. Tkfc expression is clearly stronger in duodenum than in many other mouse tissues. Therefore, I do not see a clear rationale to excludeTkfc in the sentence of line 64. Tkfc should be added to that sentence and reference 10 (that contains data for Tkfc expression —see page 358, right column, end of first paragraph; and Figure S5E) should be included in the parenthesis, together with Pubmed ID: 25409824 (Mouse ENCODE transcriptome). The author may consider deleting or substituting the word “abundantly” from line-64 sentence. If rather than mice, one considers human gene data, the conclusion would be similar.
A: Thank you very much for your suggestion. I added Tkfc to the sentence (L64) and omitted the word (abundantly).
Q2b.- In Figure 2, Tkfc is not shown as responding to ChREBP, while the contrary is stated in Figure 1 and in the text (lines 98-99 and lines 108-109; according to data in references [18,31]). Therefore, the Tkfc gene should be included in Figure 2 as being activated by ChREBP.
A2b: Thank You very much for your suggestion. Following your suggestions, I added Tkfc to the sentence (L64).
Q2c.- The paragraph devoted to Tkfc phenotypes (lines 255-272) contains several important imprecisions that should be corrected.
A2c: I corrected it.
Q3:–In line 255, DAK gene is an outdated symbol that must be substituted by TKFC, the official gene symbol assigned by the HUGO Gene Nomenclature Committee as proposed in the key paper pubmed ID 24569995. This key paper contains full data for structure and activities of triokinase/FMN cyclase, and it should be cited in line 255-256 instead of reference [71].
A3: According to your comments, I instead cited pubmed ID 24569995.
Q3: Reference [71] should be maintained in line 261, for the description of the TKFCD syndrome, just as it appears now.
A3: I maintained ref 71.
Q4: In line 258 remove “through an FMN lyase domain” since this domain name, although used in reference [71], is misleading. It implies erroneously that there is a lyase and a kinase domain. To be specific about domains, I suggest adding this description in line 258 after “…(cyclic FMN).”: “This is a homodimeric protein with subunits containing each two domains (K and L, or N-terminal and C-terminal). Both domains are needed for triokinase activity while the L domain suffices for FMN cyclase activity (Pubmed ID 24569995).”
A4: I correcte them.
Q5: Concerning the TKFCD syndrome (line 259), you should consider that it is caused by point mutations in the L (or C-terminal) domain that inactivate the triokinase activity (without data on FMN cyclase) [71]. This is important because mutations in the K (or N-terminal) domain, which also inactivate triokinase and most of FMN cyclase activity, cause a benign hair phenotype (Pubmed ID 32790068).
A5: I understand it.
Q6:–In lines 264 and 265, reference [72] should be changed to [44] (this is cited in lines 138, 140 and 141), since [44] and [72] are the same duplicated reference.
A6: thank you for your suggestion. I unitified this.
Q7: In line 269, after “Contrary to this result,” insert the following: “Thr185-TKFC is actually fully active as triokinase/FMN cyclase (Pubmed ID 24569995), and a yeast …” etc.
(Please, note as article Pubmed ID 24569995) is the only full molecular and enzymatic characterization of triokinase/FMN cyclase published and actually it is Thr185-TKFC). This is clearly stated in reference [73].
A7: Thr195-TFKC probably has almost normal activity as TKFC. Cell metabolism paper has a potent influences, Febs letters paper is important because of recently published data,.
Q3.- A more precise pattern of cites would be helpful to the reader in this part of the manuscript, to facilitate access to the original data in which the main point of the review relies.
A3: Each paper touches on gene expression, not missites, and reading only one paper makes no sense. I don't think access to each gene is that important; the reviewer's point, while understandable, is more complicated and makes the review itself harder to read. It seems strange to be so concerned with rigor there.
Q4.:The author should improve the part devoted to endogenous fructose metabolism. The relationship and differences between sections 3.1 and 3.2 are difficult to grasp. They should be better defined, with a more clear line of thought and resulting message. This could be probably improved by presenting a more structured content.
Aï¼”: I thoroughly rewrote this part.
Q5.- Gene names and symbols should be carefully revised throughout the manuscript to ensure they are shown in italics, in lower case or in capital letters as required.
A5: According to your suggestion, I correcte them.
Q6.- An abbreviation list is missing. For instance: SSBs, DHAP, G6P, X5P, TKFCD, HFrD, AR, AMPK, EPS, REE, CHO, ChoREs, MCD, Ho-1, Nqo1, TAG, etc (of course, gene symbols are not to be included in the list). In cases when the abbreviation is used only once, remove the abbreviation. Otherwise, please use them consistently throughout the manuscript.
A6: According to your suggestion. I corrected them.
MINOR ISSUES
Q7: As stated in my general comments at the beginning of this report, there are two genes which now have changed names and symbols: Glut5 is now officially Slc2a5, and Chrebp is now Mlxipl. I agree that in these cases it is better to maintain the more common old names, but the new names could be mentioned in parenthesis at their first use.
A7: According to your suggestions, I corrected it
Q8: Line 40, it reads “Fructose is metabolized faster and…”. Please state “faster than… what?
A8: According to your suggestion, I corrected it.
Q9: Figure 1A, The meaning of “Activation” in the top of the figure should be explained, modified or deleted, as necessary.
A9: I omitted “activation”.
Q10: —Line 97, it reads “…fructose regulates fructose…”. This sentence, if indeed meant, needs an explanation. However, perhaps it just should read “…GhREBP regulates fructose…” ?
A10: According to suggestion, we corrected it.
Q11: Lines 112-113, it reads “As Chrebpb is a target gene for Chrebpa,…”; should it read “As Chrebpb is a target gene for ChREBPa, …”? or else “As Chrebpb is a target gene for Chrebpa,…”? (see major point 5).
A11: According to your suggestions, I rewrote it.
Q12: Lines 286-289, I do not believe that the glycation pathway can be considered as a pathway to handle glucose effectively. It rather is a toxic consequence of high glucose levels. The sentence would be acceptable if you delete “to handle glucose effectively”.
A12: I omitted this word.
Q13: Line 293, “higher km activity” is incorrect in the spelling of KM and in the use of the term “activity”. Substitute just by “higher KM”.
A13: Thank you for your suggestion.
Q14:Iine 304, Khk-A/C, please explain that A and C are Khk isoenzymes.
A14: Thak you for your suggestion.
Q15:Line 306, please explain what are db/db mice.
A15. Db/db mice is an obese mice with leptin receptor mutation. I corrected it.
Q16:Line A14: Thnak you for your suggestion.
328, Sord should be Sord (see major point 5).
A16:I corrected it.
Q17: Line 369, substitute “activated” by “activation”.
A17: I corrected it.
Q18:Lines 374-375, the meaning of “consistent” here is unclear. Please, rephrase.
A18: I rewrote it.
Q19: Lines 399-400, this sentence needs references.
A19: This sentence was omitted.
Q20: Line 417, “since not only is fructose taken in but also excessive glucose is ingested”, mention to what situation is this referred to.
A20: When Frutcose and glucose is abosrpted at the same time, the rate of fructose uptake is increased.
Round 2
Reviewer 2 Report
The revised manuscript has been thoroughly revised by the author, and now it is adequate for publication, although a few minor points remain to be corrected.
Line 75, “AMP (adenosine triphosphate)” should be “AMP (adenosine monophosphate)”.
Line 75, “40,50” should be 4´,5´
Lines 294-302. The order of sentences in this paragraph should be changed, and one repeated sentence should be removed. Namely: the paragraph should start by the sentence of line 299, i.e. “Triokinase and FMN cyclase is an enzyme that in humans is encoded…” following until the end of the current paragraph in line 302. After this, the first lines of the current paragraph (lines 294-296) should follow, i.e. “This is a homodimeric protein… …suffices for FMN cyclase activity”. And finally the three lines repeated (lines 296-299) should be deleted.
Line 308, “bialleic” should be “biallelic”.
Line 308, “N-terminal mutation” should be “N-terminal domain”.
Lines 311-312. This sentence, from “These results are consistent… to …TKFC mutants [84]” is not true. According to reference [84], Thr185-TKFC bearing Gly192Arg mutation is fully inactive as triokinase, and conserves only about 13% of FMN cyclase activity. Thr185-TKFC bearing Arg228Trp mutation is inactive as glyceraldehyde kinase, and conserves only ≈0.2% of dihydroxyacetone kinase activity, and 8% of FMN cyclase activity. Therefore, the sentence in line 311-312 should be rephrased or just deleted.
Line 381, remove the word “activity”.
Line 536, “metabolism in lir” should be “metabolism in liver”. Pubmed record ID 5823111 reads “lir” but the original article reads “liver”.
Author Response
Q: The revised manuscript has been thoroughly revised by the author, and now it is adequate for publication, although a few minor points remain to be corrected.
A: Thank you very much for your positive comments.
Line 75, “AMP (adenosine triphosphate)” should be “AMP (adenosine monophosphate)”.
Q: Line 75, “40,50” should be 4´,5´
A: According to to your suggestions, I corrected it.
Q: Lines 294-302. The order of sentences in this paragraph should be changed, and one repeated sentence should be removed. Namely: the paragraph should start by the sentence of line 299, i.e. “Triokinase and FMN cyclase is an enzyme that in humans is encoded…” following until the end of the current paragraph in line 302. After this, the first lines of the current paragraph (lines 294-296) should follow, i.e. “This is a homodimeric protein… …suffices for FMN cyclase activity”. And finally the three lines repeated (lines 296-299) should be deleted.
A: According to your suggestions, I corrected them
Q: Line 308, “bialleic” should be “biallelic”.
A: According to your suggestions, I corrected them
Q: Line 308, “N-terminal mutation” should be “N-terminal domain”.
A: According to your suggestions, I corrected them
Q: Lines 311-312. This sentence, from “These results are consistent… to …TKFC mutants [84]” is not true. According to reference [84], Thr185-TKFC bearing Gly192Arg mutation is fully inactive as triokinase, and conserves only about 13% of FMN cyclase activity. Thr185-TKFC bearing Arg228Trp mutation is inactive as glyceraldehyde kinase, and conserves only ≈0.2% of dihydroxyacetone kinase activity, and 8% of FMN cyclase activity. Therefore, the sentence in line 311-312 should be rephrased or just deleted.
A: According to your suggestions, I deleted it.
Q: Line 381, remove the word “activity”.
A: I deleted it.
Q: Line 536, “metabolism in lir” should be “metabolism in liver”. Pubmed record ID 5823111 reads “lir” but the original article reads “liver”.
A: I corrected it.